# Mpox Vaccination Hesitancy and Its Associated Factors among Men Who Have Sex with Men in China: A National Observational Study

**DOI:** 10.3390/vaccines11091432

**Published:** 2023-08-30

**Authors:** Min Zheng, Min Du, Guanghong Yang, Yongming Yao, Xiaohan Qian, Yuan Zhi, Lin Ma, Rui Tao, Zhilin Zhu, Feng Zhou, Siqi Dai, Jie Yang, Min Liu, Jue Liu

**Affiliations:** 1Guizhou Provincial Center for Disease Control and Prevention, No.73, Bageyan Road, Yunyan District, Guiyang 550004, China; zhengmin822@sina.com (M.Z.); yaoyongming888@sohu.com (Y.Y.); qxh12272022@163.com (X.Q.); gzaidsjck@163.com (Y.Z.); mlmalin2023@163.com (L.M.); tr_47.student@sina.com (R.T.); 18311732018@139.com (Z.Z.); zf2239514005@163.com (F.Z.); daisiqi_gy@163.com (S.D.); 2The Key Laboratory of Environmental Pollution Monitoring and Disease Control, Ministry of Education, School of Public Health and Health, Guizhou Medical University, Gui’an University Town, Guiyang 550025, China; 3Department of Epidemiology and Biostatistics, School of Public Health, Peking University, No.38, Xueyuan Road, Haidian District, Beijing 100191, China; m18811579636@163.com (M.D.); liumin@bjmu.edu.cn (M.L.); 4Tianjin Shenlan Public Health Counselling Service Center, No.43, Tuanjie Ring Road, Hongqiao District, Tianjin 300122, China; tjgaga@gmail.com; 5Key Laboratory of Epidemiology of Major Diseases, Ministry of Education, Peking University, No.38, Xueyuan Road, Haidian District, Beijing 100191, China

**Keywords:** mpox, vaccination, MSM, China, nationwide

## Abstract

More than 400 confirmed mpox cases have been reported in China. The mpox vaccination is crucial to mitigate mpox transmission, especially for at-risk populations. This study aimed to determine mpox vaccination hesitancy and its associated factors in Chinese men who have sex with men (MSM). This nationwide cross-sectional study was conducted among 7538 Chinese MSM in 27 MSM social organizations from 21 provinces, municipalities, and autonomous regions of China from 31 July to 4 August 2023. Of them, the rate of mpox vaccination hesitancy was 5.59% (421/7538). The most common reason for mpox vaccination hesitation was concerns of safety and side effects (62.71%, 264/421), followed by concerns of privacy (38.24%, 161/421), thoughts of impossible infection (37.53%, 158/421), no effectiveness in preventing reinfection (30.88%, 130/421), and no worry about infection (12.35%, 52/421). Regarding the concerning characteristics of the vaccines, concerns of vaccine safety ranked first (71.74%, 5408/7538), followed by vaccine effectiveness (14.05%, 1059/7538), vaccine costs (7.35%, 554/7538), and the continuity of vaccine effectiveness (3.91%, 295/7538). The highest odds ratio of mpox vaccination hesitation was seen in MSM who were infected with mpox virus (aOR = 2.38; 95%CI = 1.08, 5.23), followed by those aged ≥60 years (aOR = 2.25; 95%CI = 1.31, 3.88), those who were unemployed (aOR = 1.66; 95%CI = 1.25, 2.19), and those who had an education level of postgraduate and above (aOR = 1.55; 95%CI = 1.01, 2.37). However, MSM who had a higher level of mpox-related knowledge (moderate: aOR = 0.53; 95%CI = 0.36, 0.77; high: aOR = 0.30; 95%CI = 0.23, 0.40) had a lower odds ratio of mpox vaccination hesitation. MSM in China had low hesitancy toward mpox vaccination. The safety and effectiveness of the vaccine and privacy were important aspects of hesitancy. Health education on mpox-related knowledge should be encouraged to promote future vaccination plans.

## 1. Introduction

Human mpox is a sporadic zoonosis caused by mpox virus (MPXV), including clade I (Central African strain or Congo Basin strain) and clade II (Western African strain) [1]. Mpox primarily occurred in rural rainforest villages of Western and Central Africa, and led to occasional exportation to other countries [2,3,4,5,6]. Since May, 2022, subsequent clusters of mpox virus infections have been reported in multiple countries [1]. The World Health Organization (WHO) Director-General declared that the mpox epidemic constituted a public health emergency of international concern (PHEIC) on 23 July 2022 [7]. As of 2 August 2023, a total of 88,600 confirmed mpox cases across 113 countries or territories have been reported globally [8]. This multicountry outbreak was driven by transmission among men who have sex with men (MSM), which was different from the mpox epidemic before 2022 [9]. During the 2022 mpox outbreak, nearly 80% of cases occurred in MSM [9]. Our previous study found that the average age of mpox cases in the 2022 human mpox outbreak was higher than that of cases before 2022 [10]. Transmission routes included skin-to-skin, oral, and rectal and perianal intimate contact, as well as possibly through seme among MSM [11]. Different from typical mpox with a widespread rash, fever, and lymphadenopathy, the clinical phenotype of human mpox now transformed into just a single or a few lesions on the genitalia, or oral and rectal mucosa [11]. Therefore, establishing more careful requirements for physical examinations and questioning about sexual history are challenges in the control of the multicountry mpox outbreak.

WHO had regular emergency meetings to assess the risk of mpox. As of 23 August 2023, WHO assessed the risk as moderate globally [12]. Although compared with other regions with moderate risk, such as the Eastern Mediterranean region, the European region and the region of the Americas, the risk was low in the Western Pacific region [12], from 1 to 31 July 2023, China reported 491 confirmed mpox cases in 23 provinces, autonomous regions, and municipalities [13]. In order to prepare for prevention and control work in advance, China announced guidelines, including the Mpox Diagnosis and Treatment Guidelines (2022 Edition) on 10 June 2022 [14], Frontier Health and Quarantine Law of the People’s Republic of China and other regulations on 24 July [15], and the Mpox prevention and control program on 26 July 2023 [16]. Considering the mobility of people and the MSM population size and activity, China should continue to take measures in the control of mpox [17]. In addition to a reduction in travel to endemic countries and risky sexual behaviors, vaccination is considered an effective way to prevent mpox infection.

WHO suggested that at-risk populations, including MSM, health workers, and laboratory personnel, should be vaccinated in the Vaccines and immunization for monkeypox: Interim guidance [18]. The development of mpox vaccines has been accelerated to mitigate mpox transmission, especially for at-risk populations. Mpox vaccines include the unavailable first-generation vaccine, second-generation vaccines like JYNNEOS (Bavarian Nordic, Hellerup, Denmark), and third-generation vaccines like MVA-BN (Modified Vaccinia AnkaraBavarian Nordic) and LC16m8 [11]. The effectiveness of the first-generation smallpox vaccine was 58% in preventing severe mpox cases [19]. Xu et al. reported that the vaccine effectiveness of MVA-BN against mpox was 87% for a one-dose vaccination and 89% for a two-dose vaccination [19]. For pre-exposure and post-exposure use in preventing mpox, the JYNNEOS vaccine, as a replication-deficient modified vaccinia Ankara, is the safest vaccine available [11]. However, vaccination hesitancy hinders the achievement of vaccine popularization. A reduction in vaccination hesitancy is essential for promoting herd immunity and improving public health emergency response capacity among MSM in China. Five cross-sectional studies have investigated mpox vaccination hesitancy in China among the general population [20], medical workers [21], male sex workers [22], MSM [23,24], and MSM living with HIV [25]. However, the abovementioned studies investigated less than 2000 participants using a convenience sampling method. In addition, there is a lack of study on the relationship between mpox virus infection and vaccination hesitancy.

Up to now, a nationwide study with a large sample size investigating vaccination hesitancy and its associated factors among MSM in China was lacking. With the assistance of the 27 MSM social organizations in China, this article provides the latest evidence on features of vaccination hesitancy, and it analyzes its associated factors to help formulate policies to achieve vaccine popularization among MSM. Additionally, considering the effect of immunosuppression on vaccination due to HIV status, we also conduct a stratified analysis on the associated factors of vaccination hesitancy by HIV status.

## 2. Materials and Methods

### 2.1. Data Source

We conducted this anonymous cross-sectional study in 27 MSM social organizations from 21 provinces, municipalities, and autonomous regions of China from 31 July to 4 August 2023 via an electronic questionnaire. The distribution of the MSM social organizations is shown in Appendix A. The inclusion criteria were the following: (1) provided informed consent; (2) identified as gay or bisexual; (3) aged ≥18 years old; (4) self-reported having sex with at least one man in the past 12 months (anal sex, oral sex, hand sex, hugging, touching, kissing, etc.); and (5) lived in mainland China. We excluded invalid questionnaires whose respondents were female or did not meet the requirements of age, sexual behavior, and residence. The study’s design, details, and procedures were approved by the Ethics Committee of the Guizhou Center for Disease Control and Prevention (Q 2022-02).

### 2.2. Measurement

According to our previous studies [24,26], a self-administered questionnaire was designed to collect information from MSM individuals, including sociodemographic characteristics, characteristics related to sexual information, and mpox-related information. The detailed questions are shown in Appendix A. We conducted a pilot survey before it was officially released. The questionnaire filling limitation was set to control quality effectively. We required health staff to guide participants to complete the questionnaire based on training content. Mpox vaccination hesitancy was our primary outcome. It was defined as the proportion of participants who answered “Both self-funded and free vaccines are not willing to take them” when asked whether they were willing to receive vaccines against monkeypox if available.

Sociodemographic characteristics included age group (18~35 years, 36~59 years, ≥60 years), occupation (employed, unemployed), ethnicity (Han, minorities), educational level (junior high school and below, high school, undergraduate, postgraduate and above), residence (eastern region, western region, central region), and marital status (married, unmarried, widowed, or divorced). Characteristics related to sexual information included sexual orientation (MSM, bisexual, or unsure), sexual disease scores (0, 1~3, 4~5), and risky sexual behavior (low, moderate, high). Mpox-related information included mpox virus infection (no, yes) and mpox-related knowledge (low, moderate, high).

Sexual diseases were assessed via three questions about HIV infection, hepatitis C infection, and other sexual diseases (syphilis, gonorrhea, condyloma acuminatum, genital herpes and genital chlamydia trachomatis, etc.). Confirmation of HIV infection and hepatitis C infection was assigned 2 points, and an answer of “unsure” was assigned 1 point; otherwise, 0 points were assigned. Confirmation of other sexual diseases was assigned 1 point; otherwise, 0 points were assigned. As a result, the sum score of the sexual disease scores ranged between 0 and 5, which were further classified into three groups (0, 1~3, 4~5). Risky sexual behavior was constructed as three sexual behaviors based on previous studies [27,28]. These sexual behaviors included condomless anal intercourse, commercial sex, and group sex in the last month. The last two behavior components were assigned 0 points when participants answered “no”; otherwise, 1 point was assigned. The behavior component of condomless anal intercourse was assigned 1 point when participants answered “always” or “often” and 0 points when participants answered “never” or “seldom”. As a result, the sum score of the risky sexual behavior index ranged between 0 and 3. High risky sexual behavior indicated that the participant engaged in all three behaviors, and low risky sexual behavior indicated that they engaged in none of the three; otherwise, moderate risky sexual behavior was indicated. Mpox virus infection was assessed via mpox symptoms and the diagnosis standard based on the Mpox prevention and control program [16]. After informing participants about the standard, they were asked “Which of the following is your case?”. Participants who answered “diagnosed by medical institution” were recognized as having an mpox virus infection. Seven items were set to assess mpox-related knowledge based on our previous studies [24,26]. Sources of mpox infection, possible routes of transmission, susceptible populations, clinical symptoms, vaccines, specific drugs, and preventive measures were all included in this section. To better quantify mpox-related knowledge, each correct answer received 1 point. The total score of mpox-related knowledge ranged from 0 to 23, and higher scores indicated better knowledge, where a score of 0 represented low mpox-related knowledge, 1 to 6 represented moderate mpox-related knowledge, and ≥7 represented high mpox-related knowledge [24,26].

### 2.3. Data Analysis

The baseline characteristics of the study population are described as means ± standard deviations (SDs) for the continuous variables and percentages for the categorical variables. The chi-square test was used in a univariate analysis. We furthermore included the significant factors in logistic regression to estimate odds ratios (ORs) with 95% confidence intervals (95% CIs). In addition, we described the reasons for vaccination hesitation and the concerning characteristics of the vaccines using bar charts.

In a sensitivity analysis, we classified those who had self-funded mpox vaccination hesitation but would accept a free mpox vaccine as having self-funded mpox vaccination hesitation to identify influencing factors. We supplemented the logistic regression to identify the influencing factors of mpox vaccination hesitation among MSM with or without HIV. All analyses were conducted using R 4.2.0. Two-sided *p* values < 0.05 were considered to be statistically significant.

## 3. Results

### 3.1. Population Characteristics

Table 1 shows the baseline characteristics of the participants. Among 7538 participants (32.20 ± 13.54 years), more than half were younger than 35 years old (72.47%), lived in eastern China (44.45%), and had at least a bachelor’s degree (51.53%).

The rate of mpox vaccination hesitation was 5.59% (421/7538). Men aged ≥60 years (14.93%), those who were unemployed (8.44%), those who were educated in junior high school and below (7.98%), those who were married (7.74%), those who were bisexual (6.89%), those who had high risky sexual behavior (10.61%) and low mpox-related knowledge (14.53%), and those who were infected with mpox virus (14.55%) (all *p* < 0.05) had a higher rate of mpox vaccination hesitation (Table 1).

We classified those who had self-funded mpox vaccination hesitation but would accept a free mpox vaccine as having self-funded mpox vaccination hesitation. The rate of self-funded mpox vaccination hesitation was 46.33% (3492/7538). The differences were also significant among age group, occupation, educational level, residence, marital status, sexual disease scores, mpox virus infection, and risky sexual behavior (all *p* < 0.05). Men aged 36–59 years (50.28%), those who were unemployed (57.81%), those who were educated in junior high school and below (54.67%), those who were widowed or divorced (55.31%), those who lived in the eastern region (48.55%), those who had high risky sexual behavior (66.67%) and low mpox-related knowledge (67.97%), and those who were infected with mpox virus (63.64%) (all *p* < 0.05) had a higher rate of mpox vaccination hesitation. But the differences were not significant among sexual orientation groups (Appendix A).

### 3.2. Reasons for Mpox Vaccination Hesitation and the Concerning Characteristics of Vaccines

Among 421 MSM with mpox vaccination hesitation, the most common reason for mpox vaccination hesitation was concerns of safety and side effects (62.71%, 264/421), followed by concerns of privacy (38.24%, 161/421), thoughts of impossible infection (37.53%, 158/421), no effectiveness in preventing reinfection (30.88%, 130/421), and no worry about infection (12.35%, 52/421) (Figure 1).

Regarding the concerning characteristics of the vaccines, concerns of vaccine safety ranked first (71.74%, 5408/7538) and was the highest, followed by vaccine effectiveness (14.05%, 1059/7538), vaccine costs (7.35%, 554/7538), and the continuity of vaccine effectiveness (3.91%, 295/7538) (Figure 2). The rate of vaccine effectiveness ranked second and was higher than 50% (Figure 2).

### 3.3. Influencing Factors of Mpox Vaccination Hesitation

Age group, occupation, educational level, marital status, sexual orientation, sexual disease scores, mpox virus infection, and risky sexual behavior were included in the logistic model. Marital status, sexual orientation, and risky sexual behavior were not associated with mpox vaccination hesitation (all *p* > 0.05) (Table 2). Men aged ≥60 years (aOR = 2.25; 95%CI = 1.31, 3.88), those who were unemployed (aOR = 1.66; 95%CI = 1.25, 2.19), those who had an education level of postgraduate and above (aOR = 1.55; 95%CI = 1.01, 2.37), and those who were infected with mpox virus (aOR = 2.38; 95%CI = 1.08, 5.23) had a higher odds ratio of mpox vaccination hesitation (all *p* < 0.05), while those who had higher mpox-related knowledge (moderate: aOR = 0.53; 95%CI = 0.36, 0.77; high: aOR = 0.30; 95%CI = 0.23, 0.40) had a lower odds ratio of mpox vaccination hesitation (both *p* < 0.05) (Table 2).

The influencing factors of self-funded mpox vaccination hesitation also included age, occupation, educational level, and mpox-related knowledge. However, men who had an education level of postgraduate and above (aOR = 0.72; 95%CI = 0.58, 0.90) had a lower odds ratio of self-funded mpox vaccination hesitation (*p* < 0.05). Additionally, men who had high risky sexual behavior (aOR = 2.01; 95%CI = 1.18, 3.43) had a higher odds ratio, while those who lived in other regions (western region: aOR = 0.88; 95%CI = 0.79, 0.97; central region: aOR = 0.86; 95%CI = 0.76, 0.98) had a lower odds ratio of self-funded mpox vaccination hesitation (Appendix A).

### 3.4. Influencing Factors of Mpox Vaccination Hesitation by HIV Status

Of 5276 MSM living without HIV, after including age group, occupation, educational level, marital status, residence, mpox virus infection, risky sexual behavior, and mpox-related knowledge in the logistic model, the influencing factors of mpox vaccination hesitation still included age, occupation, mpox virus infection, and mpox-related knowledge. Compared with all populations, educational level was not associated with mpox vaccination hesitation (*p* > 0.05), but MSM who lived in the central region (aOR = 0.84; 95%CI = 0.73, 0.98) had a lower odds ratio of mpox vaccination hesitation than MSM living without HIV (Appendix A). Of 1920 MSM living with HIV, after including age group, occupation, educational level, marital status, and mpox-related knowledge in the logistic model, the influencing factors of mpox vaccination hesitation also included age, occupation, educational level, and mpox-related knowledge. There were some differences in the influencing factors between MSM individuals living with HIV and all populations. Men aged 36–59 years old (aOR = 1.29; 95%CI = 1.04, 1.60) had a higher odds ratio of mpox vaccination hesitation (*p* < 0.05), while those with a higher educational level (undergraduate: aOR = 0.65; 95%CI = 0.48, 0.88; postgraduate and above: aOR = 0.43; 95%CI = 0.28, 0.67) had a lower odds ratio of mpox vaccination hesitation (both *p* < 0.05) (Appendix A).

## 4. Discussion

To the best of our knowledge, this is the first nationwide study investigating vaccination hesitancy and its associated factors among more than 7000 MSM in China. We found that the rate of mpox vaccination hesitation was 5.59%. We found that MSM with the highest odds of mpox vaccination hesitation were those with an mpox infection, followed by men aged ≥60 years, those who were unemployed, and those who had an education level of postgraduate and above. However, those who had higher mpox-related knowledge had a lower odds ratio of mpox vaccination hesitation.

In our study, the rate of mpox vaccination hesitation was 5.59% among MSM in China, which is lower than that in studies by Li et al. (13.85%) [23] and Zheng et al. (9.8%) [24]. The study period and investigation method are possible reasons for the differences. Firstly, these studies were conducted in 2022 when there were no local mpox cases in China. However, since July, 2023, China has reported 491 confirmed mpox cases [13]. Therefore, mpox vaccination hesitation may have alleviated due to the increased worry about infection. A previous study found that increased risk perception or concern about susceptibility to mpox infection was positively associated with vaccination intention [24,29]. Secondly, both studies used a convenience sampling method, which may have resulted in recruiting those who were more concerned about mpox. Although the rate of mpox vaccination hesitation was relatively low, it indicated that the need for mpox vaccine among MSM may be increasing. Firstly, vaccine research should be seen as an investment in the well-being of the population. Continuing to accelerate the development of mpox vaccines and promoting the mpox vaccination to high-risk populations should be encouraged.Up to now, except the unavailable first-generation vaccine, the second-generation vaccine and the third-generation smallpox vaccines were used against mpox virus. For immunocompromised populations, third-generation vaccines were safer, but it required two doses, which hindered a rapid outbreak response [11]. Secondly, it should be made aware that vaccine hesitancy still exists and that, in the context of the current mpox epidemic, it is not conducive to controlling the mpox epidemic. Therefore, understanding the reasons for vaccine hesitancy and strengthening health education to improve vaccination knowledge are necessary. Our study provides the reasons for mpox vaccination hesitation and the concerning characteristics of the vaccines. Vaccine safety was the most common reason for mpox vaccination hesitation and frequently stated concerning characteristic of the vaccines. Therefore, confirming vaccine safety and emphasizing it should be encouraged in future vaccination plans. Additionally, nearly 40% of MSM with mpox vaccination hesitation reported concerns of privacy. Due to the fear of stigma related to sexual orientation, especially in China, where same-sex relationships are not socially accepted, MSM may refuse to be vaccinated [9]. Considering concerns of privacy, anonymous vaccination may be appropriate in the future. Thirdly, the rate of self-funded mpox vaccination hesitation was higher than 45%. After assessing a variety of factors, including epidemic scale, risk population, and economic conditions, the promotion of free vaccines should be considered.

In order to help formulate more detailed vaccination policies, we found that mpox cases had the highest odds ratio of mpox vaccination hesitation, followed by older adults, those who were unemployed, and those who had a higher educational level. One study also reported similar findings regarding age [22]. Our findings supplemented more features not previously documented for the health education of key populations. However, it is noted that men who have an education level of postgraduate and above had a lower odds ratio of self-funded mpox vaccination hesitation, which may be related to the higher affordability of MSM with higher education levels. Similar results were found in MSM living with HIV. Fu et al. also reported that, compared with MSM living with HIV with a postgraduate diploma, those with a high school education or below were unwilling to get the mpox vaccine [25]. However, Zheng et al. found that the higher the education level, the lower the mpox vaccination acceptance among MSM living with HIV [24]. By exploring the influencing factors of mpox vaccination hesitation based on HIV status, the effect of education level on mpox vaccination hesitation changed. The specific reason for this difference in the relationship between education level and mpox vaccination hesitation is still unclear. More importantly, MSM who had higher mpox-related knowledge had a lower odds ratio of mpox vaccination hesitation or self-funded mpox vaccination hesitation in general MSM population. Zheng et al. also reported that vaccination acceptance was associated with a higher level of knowledge of mpox [24]. Sahin et al. found that physicians who had more information on mpox were more likely to receive the smallpox vaccine [30]. However, Winters et al. reported that almost half of the general population regarded their own knowledge level about mpox poor or very poor [29]. Additionally, even for physicians, only nearly one in three of them achieved a good level of knowledge [30]. Our previous studies found that the number of mpox cases was positively related with online search activity worldwide [31,32]. Therefore, China should promote health education on mpox-related knowledge and correct false information via multiple platforms, including Wechat, Weibo, or short-video platforms, when mpox cases are reported.

China has announced guidelines, including the Mpox Diagnosis and Treatment Guidelines (2022 Edition) [14], Frontier Health and Quarantine Law of the People’s Republic of China, and other regulations [15], to reduce importation risks before 2023. The mpox prevention and control program was announced subsequently in 26 July 2023 [16]. Although these guidelines do not mention mpox vaccination, in the Vaccines and immunization for monkeypox: Interim guidance, WHO suggested that MSM should be vaccinated [18]. Considering the present stage of local mpox cases, the mobility of people, and the MSM population size and activity, vaccination is important to mitigate the mpox epidemic [13,17]. However, there has been no recent nationwide study reporting the rate of mpox vaccination hesitation and its associated factors. In order to address these problems, our study investigated mpox virus infection, demographic characteristics, mpox-related knowledge, risky behaviors, and other sexually transmitted diseases among more than 7000 MSM from 21 provinces, municipalities, and autonomous regions in China to explore the influencing factors of vaccination hesitation.

Overall, there are some suggestions for mpox vaccination. Firstly, the current vaccine hesitancy rate is low among the Chinese MSM population, but this indicates a good time to promote mpox vaccination. Secondly, vaccine safety and effectiveness should be the focus of vaccine development, and privacy security should be the focus of the vaccination process. Finally, strengthening health education to improve mpox-related knowledge can effectively reduce vaccine hesitation, facilitating vaccination programs. In addition, it is important to emphasize that, although vaccination is effective in mitigating the mpox epidemic, a comprehensive control policy in China is still necessary. Specifically, it should include encouraging the self-screening of people from high-risk countries or those who engage in risky sexual behaviors; strengthening the training of health caregivers on the latest clinical characteristics, diagnosis, and management; and promoting comprehensive health education covering prevention, preliminary identification, isolation, etc.

The strength of this article is that it included a large sample size of more than 7000 MSM in China. However, there are some limitations to our study. First, most of the measurements were self-reported, causing recall bias. Secondly, our participants were from 27 MSM social organizations, so there was still selection bias. However, due to the hidden nature of the MSM population, a survey of the MSM population based on MSM social organizations can ensure access to the target population.

In conclusion, our national study provides the latest features of hesitancy toward mpox vaccination among MSM in China. The safety and effectiveness of the vaccines and privacy are important aspects of hesitancy. Health education on mpox-related knowledge should be encouraged to promote future vaccination plans.

## Figures and Tables

**Figure 1 vaccines-11-01432-f001:**
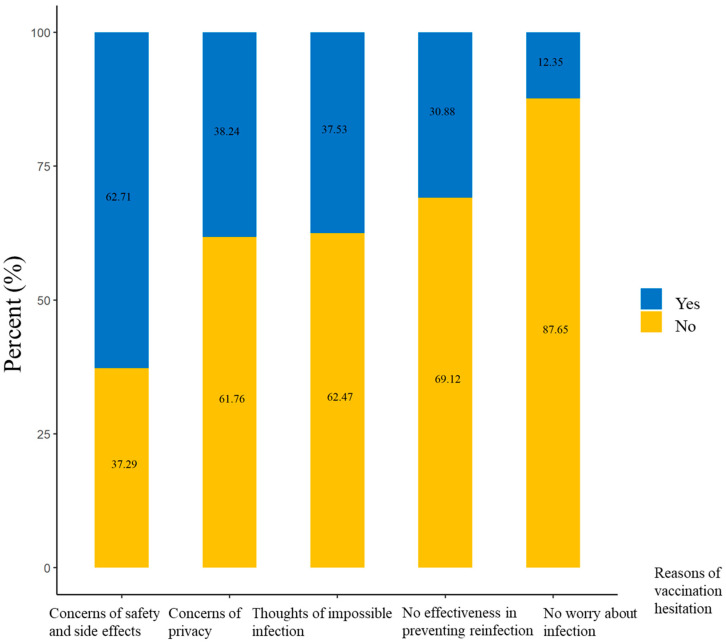
The reasons for mpox vaccination hesitation among 421 men who have sex with men and who have vaccination hesitation in China.

**Figure 2 vaccines-11-01432-f002:**
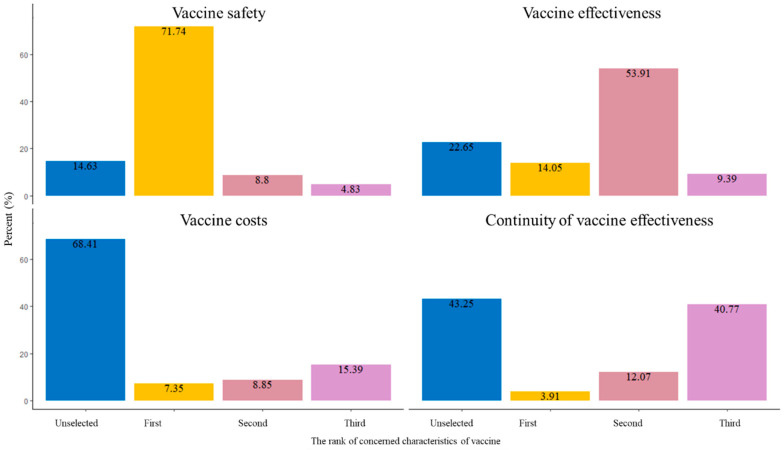
The ranks of concerning characteristics of vaccines among 7358 men who have sex with men in China.

**Table 1 vaccines-11-01432-t001:** Population characteristics by vaccination hesitation among men who have sex with men in China.

Characteristics	Overall (n, %)	Vaccination Acceptance (n, %)	Vaccination Hesitation (n, %)	*p*-Value
	7538	7117 (94.41)	421 (5.59)	
Age group (years)				<0.001
18–35	5463 (72.47)	5190 (95.00)	273 (5.00)	
36–59	1941 (25.75)	1813 (93.41)	128 (6.59)	
≥60	134 (1.78)	114 (85.07)	20 (14.93)	
Occupation				<0.001
Employed	6744 (89.47)	6390 (94.75)	354 (5.25)	
Unemployed	794 (10.53)	727 (91.56)	67 (8.44)	
Education level				0.006
Junior high school and below	664 (8.81)	611 (92.02)	53 (7.98)	
High school	2990 (39.67)	2831 (94.68)	159 (5.32)	
Undergraduate	3126 (41.47)	2970 (95.01)	156 (4.99)	
Postgraduate and above	758 (10.06)	705 (93.01)	53 (6.99)	
Marital status				0.001
Married	1008 (13.37)	930 (92.26)	78 (7.74)	
Unmarried	5993 (79.50)	5689 (94.93)	304 (5.07)	
Widowed or divorced	537 (7.12)	498 (92.74)	39 (7.26)	
Ethnicity				0.05
Han	6773 (89.85)	6407 (94.60)	366 (5.40)	
Minorities	765 (10.15)	710 (92.81)	55 (7.19)	
Residence				0.425
Eastern region	3351 (44.45)	3162 (94.36)	189 (5.64)	
Western region	2640 (35.02)	2503 (94.81)	137 (5.19)	
Central region	1547 (20.52)	1452 (93.86)	95 (6.14)	
Sexual orientation				0.006
MSM	5695 (75.55)	5401 (94.84)	294 (5.16)	
Bisexual or unsure	1843 (24.45)	1716 (93.11)	127 (6.89)	
Risky sexual behavior				0.017
Low	6003 (79.64)	5688 (94.75)	315 (5.25)	
Moderate	1469 (19.49)	1370 (93.26)	99 (6.74)	
High	66 (0.88)	59 (89.39)	7 (10.61)	
Mpox-related knowledge				<0.001
Low	537 (7.12)	459 (85.47)	78 (14.53)	
Moderate	636 (8.44)	583 (91.67)	53 (8.33)	
High	6365 (84.44)	6075 (95.44)	290 (4.56)	
Mpox virus infection				0.009
No	7483 (99.27)	7070 (94.48)	413 (5.52)	
Yes	55 (0.73)	47 (85.45)	8 (14.55)	
Sexual disease scores				0.284
0	4798 (63.65)	4517 (94.14)	281 (5.86)	
1~3	2688 (35.66)	2552 (94.94)	136 (5.06)	
4~5	52 (0.69)	48 (92.31)	4 (7.69)	

**Table 2 vaccines-11-01432-t002:** Influencing factors of mpox vaccination hesitation among 7358 men who have sex with men in China.

Characteristics	aOR (95% CI)	*p*-Value
Age group (years)		
18–35	1 (reference)	
36–59	1.20 (0.93, 1.55)	0.160
≥60	2.25 (1.31, 3.88)	0.003
Occupation		
Employed	1 (reference)	
Unemployed	1.66 (1.25, 2.19)	<0.001
Education level		
Junior high school and below	1 (reference)	
High school	0.91 (0.65, 1.28)	0.592
Undergraduate	1.03 (0.72, 1.46)	0.887
Postgraduate and above	1.55 (1.01, 2.37)	0.043
Marital status		
Married	1 (reference)	
Unmarried	0.81 (0.59, 1.10)	0.173
Widowed or divorced	0.94 (0.62, 1.42)	0.773
Sexual orientation		
MSM	1 (reference)	
Bisexual or unsure	1.21 (0.96, 1.53)	0.106
Risky sexual behavior		
Low	1 (reference)	
Moderate	1.17 (0.92, 1.50)	0.195
High	1.40 (0.60, 3.22)	0.435
Mpox-related knowledge		
Low	1 (reference)	
Moderate	0.53 (0.36, 0.77)	<0.0001
High	0.30 (0.23, 0.40)	<0.0001
Mpox virus infection		
No	1 (reference)	
Yes	2.38 (1.08, 5.23)	0.031

Notes: OR = odds ratio; 95%CI = 95% confidence interval.

## Data Availability

Data are available according to the corresponding author’s permission.

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
