# Peer review of "Mpox Vaccination Hesitancy and Its Associated Factors among Men Who Have Sex with Men in China: A National Observational Study"

_vaccines, 2023, doi:10.3390/vaccines11091432_

Round 1
Reviewer 1 Report
I recommand the publication of this study findings, with few minor improvements.
1) The number of MSM related organizations involved : 25 or more as mentionned at the end of the article ?
2) The discussion section has to be reviewed to avoid repetitions, which are redundant
3) For the sake of clarity, in the abstract and discussion section, about the influencing epidemiological factors ( age, education, employement....),it should be made clear whether those with highest odds are :
- MEN aged above 60 years with higher education , employement..... or
- The following population groups such those aged above 60,
those with higher education,
those who are employed,
those who are....
It is clear in the findings presentation with statistix figures.
4) I suggest to give more information in the methodology section on participant selection, on the MSM centers/ organization which distributed and collecteLesd the questionnaires to the participants. This information detail is important as this may be the only study methodology limit.
This is a good study for international publication delivery.
few easy to do editing improvements .
( see above my suggestions)
Author Response
Dear Editors and Reviewers:
Thank you for your letter and for the reviewers’ comments concerning our manuscript entitled “Mpox vaccination hesitancy and its associated factors among men who have sex with men in China: A national observational study” (Submission ID vaccines-2589314). Those comments are all valuable and very helpful for revising and improving our paper. We have made the requested changes which we hope meets your approval. The changes were noted in red in the revised version. The main modification in the paper and the responds to the reviewers and editors’ comments are as following:
Reviewer(s)' Comments to Author:
Reviewer: 1
I recommand the publication of this study findings, with few minor improvements.
Response: Thanks.
1) The number of MSM related organizations involved : 25 or more as mentionned at the end of the article ?
Response: Thanks. We have corrected the number of MSM related organizations as 27 in full article.
2) The discussion section has to be reviewed to avoid repetitions, which are redundant
Response: Thanks. We have reviewed discussion to avoid repetitions and redundancy.
3) For the sake of clarity, in the abstract and discussion section, about the influencing epidemiological factors ( age, education, employement....),it should be made clear whether those with highest odds are :
- MEN aged above 60 years with higher education , employement..... or
- The following population groups such those aged above 60, those with higher education,those who are employed,those who are....
It is clear in the findings presentation with statistix figures.
Response: Thanks. We have represented the abstract and discussion section to point those with highest odds as shown in page 1 line 36-42 : “The highest odds ratio of mpox vaccination hesitation was seen in MSM who were infected with mpox virus (aOR=2.38; 95%CI=1.08, 5.23), following by those aged≥60 years (aOR=2.25; 95%CI=1.31, 3.88), those were unemployed (aOR=1.66; 95%CI=1.25, 2.19), and those who educated in postgraduate and above (aOR=1.55; 95%CI=1.01, 2.37). However, MSM who had higher level of mpox-related knowledge (moderate: aOR=0.53; 95%CI=0.36, 0.77; high: aOR=0.30; 95%CI=0.23, 0.40) had lower odds ratio of mpox vaccination hesitation.” and page 7 line 214-217 “We found that MSM with highest odds of mpox vaccination hesitation are mpox cases, following by men aged≥60 years, those who were unemployed, and those who edu-cated in postgraduate and above. However, those who had higher mpox-related knowledge had lower odds ratio of mpox vaccination hesitation.”
4) I suggest to give more information in the methodology section on participant selection, on the MSM centers/ organization which distributed and collecteLesd the questionnaires to the participants. This information detail is important as this may be the only study methodology limit.
Response: Thanks. As for the distribution of MSM centers/ organizations of these study, we have added the information in the supplemental table 1. We required health staffs to guide participants to complete the questionnaire based on training content. We have supplemented more information in the methodology section on participant selection as shown in page 2 line 87-88 “The distribution of MSM social organizations was seen in Figure S1 and Table S1.” and page 3 line 97-102 “According to our previous studies [22, 24], a self-administered questionnaire was designed to collect information from the MSM individuals, including sociodemo-graphic characteristics, characteristics related to sexual information and mpox related information (Table S2). We conducted a pilot survey before it was officially released. The questionnaire fill-ing limitation was set to control quality effectively. We required health staffs to guide participants to complete the questionnaire based on training content.”.
This is a good study for international publication delivery.
Response: Thanks.
Reviewer 2 Report
The evaluated article presents a national observational study that investigates the hesitancy towards mpox vaccination and its associated factors among men who have sex with men (MSM) in China. The study surveyed over 7,000 participants from various regions of China and examined their attitudes, reasons for hesitation, and concerns related to the mpox vaccine. The study aims to contribute valuable insights into promoting vaccination plans for MSM, particularly in light of the emerging mpox epidemic. Study design and presentation of the results are quite appropriate for the scope of the manuscript. However, there are some issues and questions that should be addressed in the study.
1- Please indicate HIV status and whether there were any differences in the analyses among individuals MSM with HIV.
2-Exclusion criteria should be added to the methodology section.
3- They did not mention how the data collection tool and questionnaire were developed and if they test its validity.
4- It would be better to add a discussion about the general population's (https://doi.org/10.1371/journal.pone.0278622) and doctors' (https://doi.org/10.3390/vaccines11010019) knowledge levels and attitudes regarding vaccines.
5-Please provide the entire survey as a supplement.
Moderate editing of English language required
Author Response
Dear Editors and Reviewers:
Thank you for your letter and for the reviewers’ comments concerning our manuscript entitled “Mpox vaccination hesitancy and its associated factors among men who have sex with men in China: A national observational study” (Submission ID vaccines-2589314). Those comments are all valuable and very helpful for revising and improving our paper. We have made the requested changes which we hope meets your approval. The changes were noted in red in the revised version. The main modification in the paper and the responds to the reviewers and editors’ comments are as following:
Reviewer: 2
The evaluated article presents a national observational study that investigates the hesitancy towards mpox vaccination and its associated factors among men who have sex with men (MSM) in China. The study surveyed over 7,000 participants from various regions of China and examined their attitudes, reasons for hesitation, and concerns related to the mpox vaccine. The study aims to contribute valuable insights into promoting vaccination plans for MSM, particularly in light of the emerging mpox epidemic. Study design and presentation of the results are quite appropriate for the scope of the manuscript. However, there are some issues and questions that should be addressed in the study.
1- Please indicate HIV status and whether there were any differences in the analyses among individuals MSM with HIV.
Response: Thanks. We have added the results of analyses among individuals MSM with HIV exclusion criteria as shown in page 4 line 148-150: “We supplemented logistic regression to find the influencing factors of mpox vaccina-tion hesitation among MSM with HIV.” and page 7 line 202-210: “Of 1920 MSM with HIV, after included age group, occupation, educational level, mar-ital status, and mpox-related knowledge in logistic model, the influencing factors of mpox vaccination hesitation also included age, occupation, educational level and mpox-related knowledge. There were some differences in the analyses among indi-viduals MSM with HIV and all population. Men aged 36-59 years old (aOR=1.29; 95%CI=1.04, 1.60) had higher odds ratio of mpox vaccination hesitation (P<0.05), while those with higher educational level (undergraduate: aOR=0.65; 95%CI=0.48, 0.88; postgraduate and above: aOR=0.43; 95%CI=0.28, 0.67) had low-er odds ratio of mpox vaccination hesitation (both P<0.05) (Table S5).”.
2-Exclusion criteria should be added to the methodology section.
Response: Thanks. we have added the exclusion criteria as shown in page 2 line 91-93“We excluded invalid questionnaires, whose respondents were female or did not meet the requirements of age, sexual behavior, and residence.”.
3- They did not mention how the data collection tool and questionnaire were developed and if they test its validity.
Response: Thanks. We have supplemented more information in the methodology section on participant selection as shown in page 3 line 97-100“According to our previous studies [22, 24], a self-administered questionnaire was designed to collect information from the MSM individuals, including sociodemo-graphic characteristics, characteristics related to sexual information and mpox related information (Table S2). We conducted a pilot survey before it was officially released. The questionnaire fill-ing limitation was set to control quality effectively. We required health staffs to guide participants to complete the questionnaire based on training content.”.
4- It would be better to add a discussion about the general population's (https://doi.org/10.1371/journal.pone.0278622) and doctors' (https://doi.org/10.3390/vaccines11010019IF: 7.8 Q1 ) knowledge levels and attitudes regarding vaccines.
Response: Thanks. We have added a discussion about the general population's and doctors' knowledge levels and attitudes regarding vaccines as shown in page 7 line 224-226“Previous study found that increased risk perception or concerns about susceptibility to mpox infection was positively associated with vaccination intentions [22, 27].” and page 8 line 241-248“Furthermore, MSM who had higher mpox-related knowledge had lower odds ratio of mpox vaccination hesitation. Zheng et al. also reported vaccination acceptance was as-sociated with higher level of knowledge of mpox [22]. Sahin et al. found that physi-cians who had more information on mpox were more likely to receive the smallpox vaccine [28]. However, Winters et al. reported that almost half the general population regarded their own knowledge level about mpox poor or very poor [27]. Besides, even for physicians, only nearly one in three of them achieved a good level of knowledge [28].”.
5-Please provide the entire survey as a supplement.
Response: Thanks. We have added the survey questionnaire in Table S2.
Round 2
Reviewer 2 Report
I am satisfied that the authors have addressed all of my previous concerns about the article. It is now much improved and I feel that it is now suitable for publication.
Minor editing of English language required